# Scalable and programmable topological transitions in plasmonic Moiré superlattices

Bo Tian [1], Xi Zhang [2], Ruitao Wu[1], Yuquan Zhang[1], Luping Du [1] ✉ & Xiaocong Yuan [1]

Topological transitions are fundamental phenomena in electronics, photonics, and quantum technologies. However, their scalability and tunability are constrained by material properties or structural rigidities. Here, we demonstrate that plasmonic Moiré superlattices offer a platform for programmable, large-range topological transitions via wavefront engineering. By tailoring the phases of elementary evanescent waves in hexagonal systems, we create Moiré lattices of optical skyrmions, whose topological invariants evolution is programmable and scalable. Theoretical calculations indicate that the topological invariants span a range of values going from −58 to +58, extendable by tuning the Moiré angle. Remarkably, these values are constrained by symmetry to exclude integer multiples of 3/2, revealing an intrinsic link between symmetry and topological quantization. Our work establishes a versatile control platform of real-space topology for exploring topological transitions mechanisms and studying critical topological phenomena, further promoting breakthroughs in structured light, photonic computing, and condensed matter physics.

Topological transitions (TTs) characterize the transformations between distinct topological states[1–3], playing a fundamental role in electronics[4–6], photonics[7–9], and quantum systems[10]. Conventionally, TTs have been extensively studied in terms of band structures in momentum-space[4–9,11,12]. Topological band theory describing the topological phase transitions of band structures advances theoretical physics and sheds light on edge states in topological materials, providing a foundation for low-energy electronic and photonic devices[2,13]. Related studies have also been focused on the topological structures in real-space, especially the well-known skyrmions, which open new avenues for high-density information storage, quantum computing, and structured light control[14–29]. However, in both scenarios, achieving scalable and programmable TTs remains a crucial challenge, as these systems are constrained by their reliance on intrinsic material properties or predefined architectures. The fundamentally limited dynamic tunability of the system has been the main challenge for advancing TT's theory and further comprehensive studies of critical topological phenomena.

In Moiré physics, Moiré superlattices, formed by the relative displacement or rotation of periodic structures, introduce a highly tunable degree of freedom that enables extensive structural modulation[30–32]. Initially explored in twisted bilayer graphene[33], Moiré physics has unveiled several groundbreaking discoveries, including unconventional superconductivity[33–35], the quantum anomalous Hall effect[36,37], and controllable magnetism[38]. Subsequent advances have expanded Moiré platforms to plasmonic[39–41], hydrodynamics[42], and acoustic systems[43], which exhibit exotic dispersion relations[39], nontrivial vector field structures[40–42], and skyrmion-like topological features[40,41]. These discoveries underscore their potential for applications in energy transfer, particle navigation, high-efficiency light manipulation and reconfigurable optical devices. In particular, the Moiré superlattices generated in plasmonic platforms[40,41] can support arbitrarily large topological invariants. This capability not only markedly enriches the diversity of topological states but also facilitates topological transitions between these states.

[1]Nanophotonics Research Centre, Shenzhen Key Laboratory of Micro-Scale Optical Information Technology, Institute of Microscale Optoelectronics & State Key Laboratory of Radio Frequency Heterogeneous Integration, Shenzhen University, Shenzhen, China. [2]Guangdong Provincial Key Laboratory of Micro/Nano Optomechatronics Engineering, College of Mechatronics and Control Engineering, Shenzhen University, Shenzhen, China. ✉e-mail: lpdu@szu.edu.cn

Here, we introduce plasmonic Moiré superlattices as a programmable platform for achieving extensive TTs of optical skyrmions. By precisely engineering the elementary evanescent waves in a hexagonal optical system, we dynamically reconfigure both the skyrmion lattice and its associated topological state. More importantly, when two such lattices are overlaid to form a Moiré superlattice, the system exhibits programmable, symmetry-governed topological control, enabling unprecedented expansion of the topological space. Theoretical calculations indicate that TTs occur between skyrmion configurations carrying integer and half-integer topological invariants (TIs) within the range of −8 to +8, excluding TIs that are integer multiples of 3/2 at a Moiré angle of 13.17°. The accessible range of TIs further expands to −19 to +19 at 9.43°, and reaches −58 to +58 at 25.04°, representing one of the largest-scale TTs reported to date in physical systems. This selection rule, governed by the discrete symmetry of the lattice, reflects an intrinsic constraint on real-space topological quantization. Experimentally, we investigated the evolution of the topological states of the vector field in the Moiré superlattice at a Moiré angle of 13.17°. Although measurements were conducted at this single angle, the experimental results clearly confirm the occurrence of the predicted topological transitions, consistent with the theoretical calculations regarding the presence and distribution of these topological states. Moreover, the connection between the TTs of skyrmions and the topological phase transitions of energy bands has also been identified. Our work has not only provided a deep insight into the universality of TTs across various physical systems but also established an extensive and programmable topological photonic platform, which unlocks new opportunities for structured light control, topological photonic computing, and high-dimensional information processing.

## Results

### Formation and mechanism of optical TTs

Skyrmions are topological structures whose invariants count the number of times a vector field warps the unit sphere[20]. This topology has been established in optics to characterize the intrinsic topological properties of vector fields[22-29]. In two-dimensional real-space systems, the TI of a vector field can be quantified by ref. [22]:

$$TI = \frac{1}{4\pi} \int_D \mathbf{n} \cdot \left( \partial_x \mathbf{n} \times \partial_y \mathbf{n} \right) dxdy, \tag{1}$$

where $\mathbf{n}$ is a normalized vector, $TI$ represents the number of times the normalized vector field wraps around the unit sphere ($S^2$) as the 2D spatial domain is traversed. In optical systems, the unit vector $\mathbf{n}$ can be constructed from either the electric ($\mathbf{E}$) or magnetic field ($\mathbf{H}$), yielding field skyrmions[22], or from the spin density ($\mathbf{S}$), leading to spin skyrmions[23]. In this work, to induce the TTs of field skyrmions, we synthesize the underlying electric field by tailoring the interference of six transverse magnetic (TM) evanescent modes, enabling precise control over the emerging topological textures. The out-of-plane electric field component of the vector field is given by (Supplementary Note 1):

$$E_z = E_0 \sum_{\alpha=1}^3 \cos\left(\mathbf{k}_{s,\alpha} \cdot \mathbf{r} + \varphi_\alpha\right), \tag{2}$$

where $E_0$ denotes the amplitude, $\mathbf{k}_{s,\alpha}$ is the in-plane wavevector, $\varphi_\alpha$ are the phases, and the relations among $\mathbf{k}_{s,\alpha}$ and $\varphi_\alpha$ are illustrated in Fig. 1a. The in-plane field components, $E_x$ and $E_y$, can be obtained with known $E_z$ through Maxwell's equations (Supplementary Note 1).

To characterize the influence of the phases $\varphi_\alpha$ on the resulting vector field described by Eq. (2), we introduce a synthetic phase parameter $\beta = \varphi_2 - \varphi_1 - \varphi_3$, which not only reduces the dimensionality of parameter regulation but also encompasses all possible scenarios (Supplementary Note 1). In the numerical implementation, the

relationship between the phase and $\beta$ can be defined as $\varphi_2 = \beta/3$ and $\varphi_1 = \varphi_3 = -\beta/3$. This choice not only better reflects the symmetry of the structure but also ensures that the center of the unit cell remains at the coordinate origin. The spatial distributions of $E_z$ for $\beta = 0$ and $\beta = \pi/2$ are presented in Fig. 1b, c, respectively, accompanied by the corresponding normalized in-plane electric field vectors. These isolated topological states were previously obtained through methods equivalent to phase modulation[44]. Additional $E_z$ field distributions for other $\beta$ values are provided in Supplementary Fig. 1. The electric fields form two-dimensional hexagonal lattices, with unit cells outlined by black dashed lines. The TI, computed via Eq. (1), is found to be 1 for $\beta = 0$ and 0 for $\beta = \pi/2$. This observation indicates that the variations of $\beta$ alter not only the morphology but also the topological state of the vector field. To systematically explore the dependence of TI on $\beta$, we calculated the TI for $\beta$ values over a large range as plotted in Fig. 1d (dashed line). This relation reveals a periodicity of $2\pi$, within which three distinct topological states emerge, characterized by TIs of 1, 0, and −1. Specifically, the TI remains unchanged as $\beta$ varies within several open intervals, such as (−π/2, π/2) and (π/2, 3π/2), indicating the robustness of the topological state. Besides, a discrete jump in the TI occurs as $\beta$ crosses π/2, signifying a TT in the vector field configuration.

The mechanism underlying the TT of optical skyrmions mirrors that underlying topological phase transitions in topological insulators −both are driven by the emergence of singularities in the system. This commonality is elucidated by analyzing the topological invariants of skyrmions and comparing them to those arising in the two-band model used for topological insulators. Within topological insulators, the two-band model, which captures the evolution of the band structure across topological phase transitions, features energy bands of the form ±|$\mathbf{h}$|. The Chern number characterizing the topological nature of the band structure is given by a formula identical to that in Eq. (1). Here, $\mathbf{h}$ denotes a vector in the Brillouin zone parameterized by the hopping strengths. Consequently, the Chern number can be interpreted as the topological invariant of a skyrmion by treating $\mathbf{h}$ as a mapping from the first Brillouin zone to the real-space sphere $S^2$. In topological insulators, a topological phase transition is triggered when the energy bands close and reopen. Band closure implies that the vector $\mathbf{h}$ must vanish ($\mathbf{h} = 0$) at one or more points in the Brillouin zone. By extending this concept to vector fields defined on a hexagonal lattice, we deduce that a necessary condition for a TT in such a system is similarly the emergence of singularities within the vector field. A detailed theoretical treatment is provided in Supplementary Note 2.

To verify the analogous role of singularities in the optical skyrmion system, we define the parameter $E_1 = |\mathbf{E}|_{min}$, representing the minimum absolute value of the vector field. The dependence of $E_1$ on the phase parameter $\beta$ is shown in Fig. 1d (solid line), which reveals that topological state transitions occur precisely at points where $E_1 = 0$. This confirms that the emergence of a singularity is the necessary condition for a TT. In the absence of such singularities, the TI remains unchanged, despite continuous deformations of the vector field. To further reinforce this analogy, we define an energy band-like structure based on the commonality of Eq. (1) and the Chern number in the two-band model. This structure takes the form ±|$\mathbf{E}$|, exhibiting a well-defined gap for $\beta = 0$ or π (Fig. 1e) and a gapless configuration at $\beta = \pi/2$ (Fig. 1f). This gap closing coincides with an abrupt TT. Upon further increasing $\beta$, the gap reopens, and the system enters a different topological phase. This behavior mimics the gap-closing and reopening process in conventional topological band transitions, providing a clear real-space analogy for topological phase transitions.

### Optical TT in Moiré superlattices

The properties of Moiré superlattices formed by interlayer twisting differ markedly from those of single-layer lattices[43]. The formation of a hexagonal Moiré superlattice (Fig. 2a) entails the interference of two

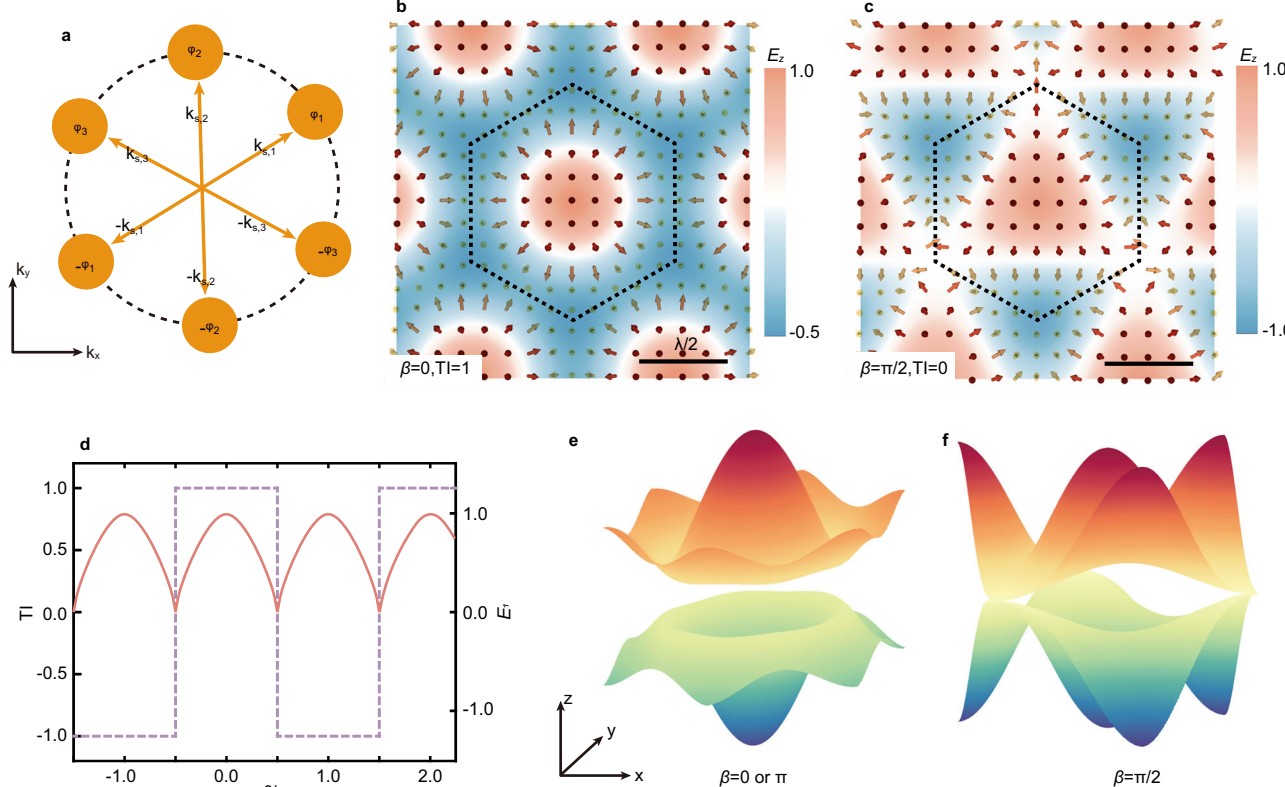

**Fig. 1 | Formation and mechanism of optical topological transition. a** Schematic illustration of the relation among wave vectors and phases of the six counter-propagating evanescent plane waves described by Eq. (2). Distribution of the out-of-plane electric field component $E_z$ when $\beta = 0$ (**b**) and $\beta = \pi/2$ (**c**), showing a two-dimensional hexagonal lattice. The normalized electric field vectors reveal the formation of a topological structure with a topological invariant (TI) of 1 (**b**) and 0 (**c**). **d** The dashed line shows the trend of TI as a function of $\beta = \varphi_2 - \varphi_1 - \varphi_3$. Three distinct topological states with TIs of 1, 0, and −1 are presented. The evolution of $E_1 = |\mathbf{E}|_{\min}$ as a function of $\beta$ is also shown by a solid line, illustrating the periodic occurrence of the singularities ($\mathbf{E} = \mathbf{0}$), corresponding to the occurrence of topological transitions of optical skyrmion. Energy band-like structure for $\beta = 0$ or $\pi$ (**e**) and $\beta = \pi/2$ (**f**). During the TTs, the skyrmions evolution is analogous to that of energy bands. Consequently, the energy band-like structure undergoes a change consistent with that of energy bands: it transitions from a gapped structure (**e**) to a gapless structure (**f**) and back to a gapped structure (**e**).

identical hexagonal lattices with a relative angle $\theta$ to generate a larger-scale hexagonal Moiré pattern. Given that the vector field described by Eq. (2) exhibits intrinsic hexagonal lattice symmetry, an optical Moiré superlattice can be constructed by coherently superimposing these two identical vector fields with a mutual rotation angle $\theta$. The resulting out-of-plane electric field component can be expressed as:

$$E_z = E_0 \sum_{\alpha=1}^{3} \left[ \cos(\mathbf{k}_{s,\alpha,1} \cdot \mathbf{r} + \varphi_{\alpha,1}) + \cos(\mathbf{k}_{s,\alpha,2} \cdot \mathbf{r} + \varphi_{\alpha,2}) \right] \quad (3)$$

where the subscripts 1 and 2 distinguish the two hexagonal vector fields. The relation among the in-plane wavevectors $\mathbf{k}_{s,\alpha,1}$ and $\mathbf{k}_{s,\alpha,2}$, as well as the phase terms $\varphi_{\alpha,1}$ and $\varphi_{\alpha,2}$, are shown schematically in Fig. 2b. Further details regarding the construction and interpretation of the Moiré vector fields can be found in Supplementary Note 3.

We introduce two synthetic phase parameters, $\beta_1 = \varphi_{2,1} - \varphi_{1,1} - \varphi_{3,1}$, and $\beta_2 = \varphi_{2,2} - \varphi_{1,2} - \varphi_{3,2}$, to conveniently quantify the influence of the phase configurations on the resulting Moiré vector field. As we mentioned previously, each of these parameters can capture the essential phase modulation within each component vector field. Within the computational framework, the phases are related to the parameters $(\beta_1, \beta_2)$ according to the convention of the hexagonal lattice and are selected as follows: $\varphi_{2,1} = \beta_1/3$, $\varphi_{1,1} = \varphi_{3,1} = -\beta_1/3$, $\varphi_{2,2} = \beta_2/3$, and $\varphi_{1,2} = \varphi_{3,2} = -\beta_2/3$. For the hexagonal Moiré superlattice, the Moiré angle $\theta$ satisfies the relation: $\cos\theta = \left[(n+m)^2 + 2mn\right] / \left[2(n+m)^2 - 2mn\right]$, where $m$ and $n$ are positive integers[45]. The vector fields corresponding to $m = 2$ and $n = 3$ are

presented for $\beta_1 = \beta_2 = 0$ (Fig. 2c) and $\beta_1 = \beta_2 = \pi$ (Fig. 2d), with additional configurations presented in Supplementary Fig. 2. In all cases, black dashed lines indicate the unit cells of the hexagonal super-lattices. The calculated TIs for these configurations are 7 (Fig. 2c) and −8 (Fig. 2d), indicating that the phase parameters $\beta_1$ and $\beta_2$ in the Moiré vector field described by Eq. (3) play a role analogous to the single $\beta$ parameter in Eq. (2)−both modulate the topological states of the vector field. The dependence of TI on both $\beta_1$ and $\beta_2$ is shown in Fig. 2e. The parameter space is partitioned and color-coded into multiple regions with associated TI. Within the range from −8 to 8, the TIs span all integers and half-integers, except for $\pm 1.5$, $\pm 3$, $\pm 4.5$ and $\pm 6$. Notably, 0 and half-integer values are not prominently visible in Fig. 2e because they lie on the boundaries between adjacent regions. At these boundaries, the TIs assume the average value of the neighboring regions' TIs (Supplementary Note 4). These results indicate that the topological state remains unchanged as long as $\beta_1$ and $\beta_2$ vary within a given region. However, a discrete TT occurs whenever $(\beta_1, \beta_2)$ cross from one region to another, resulting in abrupt changes in the TI within the range of −8 to 8.

As we demonstrated in Fig. 1, singularities within the vector field serve as indispensable prerequisites for the TT. Consequently, the boundaries of the regions in Fig. 2e correspond to the zero contours of the vector field described by Eq. (3). The values of $E_1 = |\mathbf{E}|_{\min}$ as a function of $\beta_1$ and $\beta_2$ are presented in Fig. 2f. From the comparison between Fig. 2e, f one can notice that the region boundaries and the zero trajectories are perfectly aligned. Notably, at positions marked by black dashed lines in Fig. 2e, where

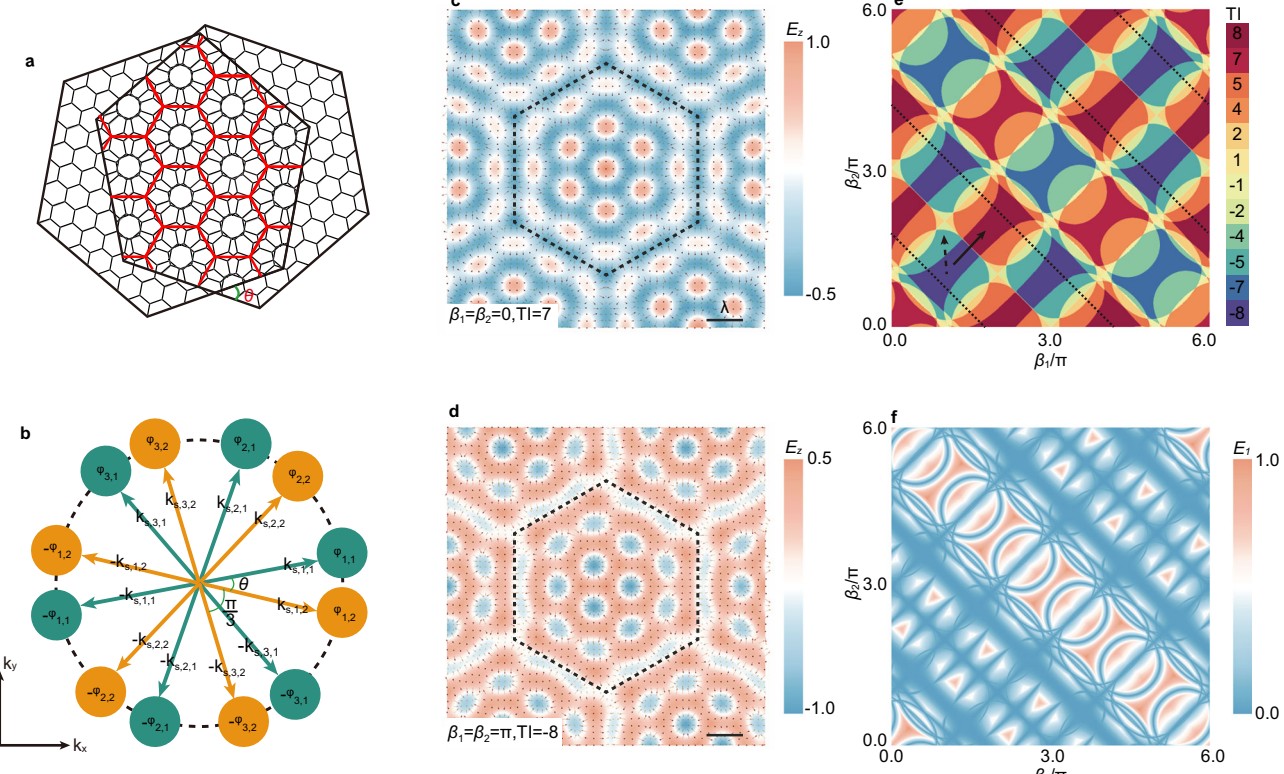

**Fig. 2 | Large-range optical topological transition in Moiré superlattices.**
**a** Schematic illustration of the interference between two hexagonal lattices with a relative twist angle $\theta$, forming a hexagonal Moiré superlattice with a larger unit cell (marked by red lines). **b** Schematic illustration of the wave vectors and phases of the twelve counter-propagating evanescent plane waves described by Eq. (3). Distribution of the out-of-plane electric field component $E_z$ when $\beta_1 = \varphi_{2,1} - \varphi_{1,1} - \varphi_{3,1}$ and $\beta_2 = \varphi_{2,2} - \varphi_{1,2} - \varphi_{3,2}$ simultaneously are 0 (**c**) and $\pi$ (**d**), showing a hexagonal superlattice with topological invariants of 7 (**c**) and $-8$ (**d**), respectively. The normalized electric field vectors at each location are indicated by arrows. **e** The TIs

when $\beta_1$ and $\beta_2$ change independently. The pattern is divided into several small areas, each with a definite TI, which assume all integer and half-integer values in the interval from $-8$ to 8, except for $\pm 1.5$, $\pm 3$, $\pm 4.5$ and $\pm 6$. Values of 0 and half-integers all lie on the boundaries of each small area, and the TIs on the boundaries equal the average of the TIs in the small areas on either side of the boundaries. **f** The minimum value of the vector field described by Eq. (3) varies with $\beta_1$ and $\beta_2$. The minimum equal to 0 implies the existence of at least one singularity. **e**, **f** show that the trajectory of the singularity is the same as the trajectory of the topological transition.

singularities also appear, the TIs on either side remain unchanged. This underscores that the mere occurrence of singularities is a necessary yet insufficient criteria for triggering a TT. The transition requires not only that the field vanishes but that the vector field undergoes a topological reconfiguration in its local orientation. A more detailed analysis is provided in the Supplementary Note 4.

While the TI range is confined to $-8$ to 8 for the parameter set $(m, n) = (2, 3)$, as illustrated in Fig. 2, the accessible TI range expands dramatically with increasing $m$ and $n$. For example, when $(m, n) = (3, 4)$, the range extends from $-19$ to 19; and for $(m, n) = (4, 9)$, it further upscales from $-58$ to 58. This scaling behavior highlights the ability of Moiré structuring to support a broad range of topological states (Supplementary Note 5).

### Symmetry-constrained discretization of TIs

An interesting yet crucial finding in Fig. 2e is that the TIs never take-values that are integer multiples of $3/2$. This phenomenon arises from the distribution and classification of singularities, as well as the $C_3$ symmetry of the structure. The Moiré superlattice described by Eq. (3) possesses $C_3$ symmetry, where symmetric points are located either at the center or the vertices of the unit cell. Therefore, the center and vertices of the unit cell are termed high-symmetry points, while other positions are termed general-position points. Accordingly, singularities located at high-symmetry points are classified as high-symmetry singularities (red circles in Fig. 3a), whereas those occurring at general

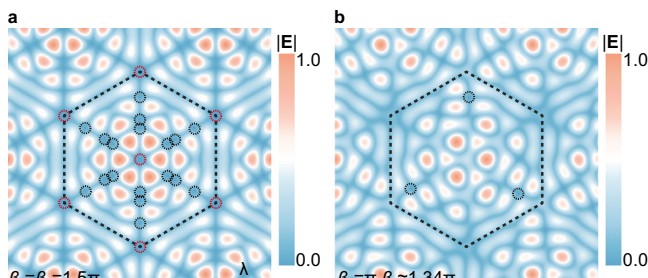

**Fig. 3 | Symmetry-constrained discretization of topological invariants.** The unit cells with (**a**) and without (**b**) high-symmetry singularities. High-symmetry singularities (highlighted with red circles) lead to reversal of topological invariants (TIs). Due to the antisymmetric nature of the vector field with respect to the phase parameter, TIs go from $-8$ to 8, as shown with the solid arrow in Fig. 2e. Generic-position singularities (highlighted with black circles) lead to incremental changes of the TI by integer multiples of 3 (for example, varying from $-8$ to $-5$, as shown with the dashed arrow in Fig. 2e), as dictated by the local winding number of the field near each singularity and the global $C_3$ symmetry constraint.

positions are regarded as general-position singularities (black circles in Fig. 3a, b).

When high-symmetry singularities are present, they always force TI sign reversal (e.g., from $-8$ to 8, as shown with the solid arrow in Fig. 2e) whether or not generic-position singularities coexist. This

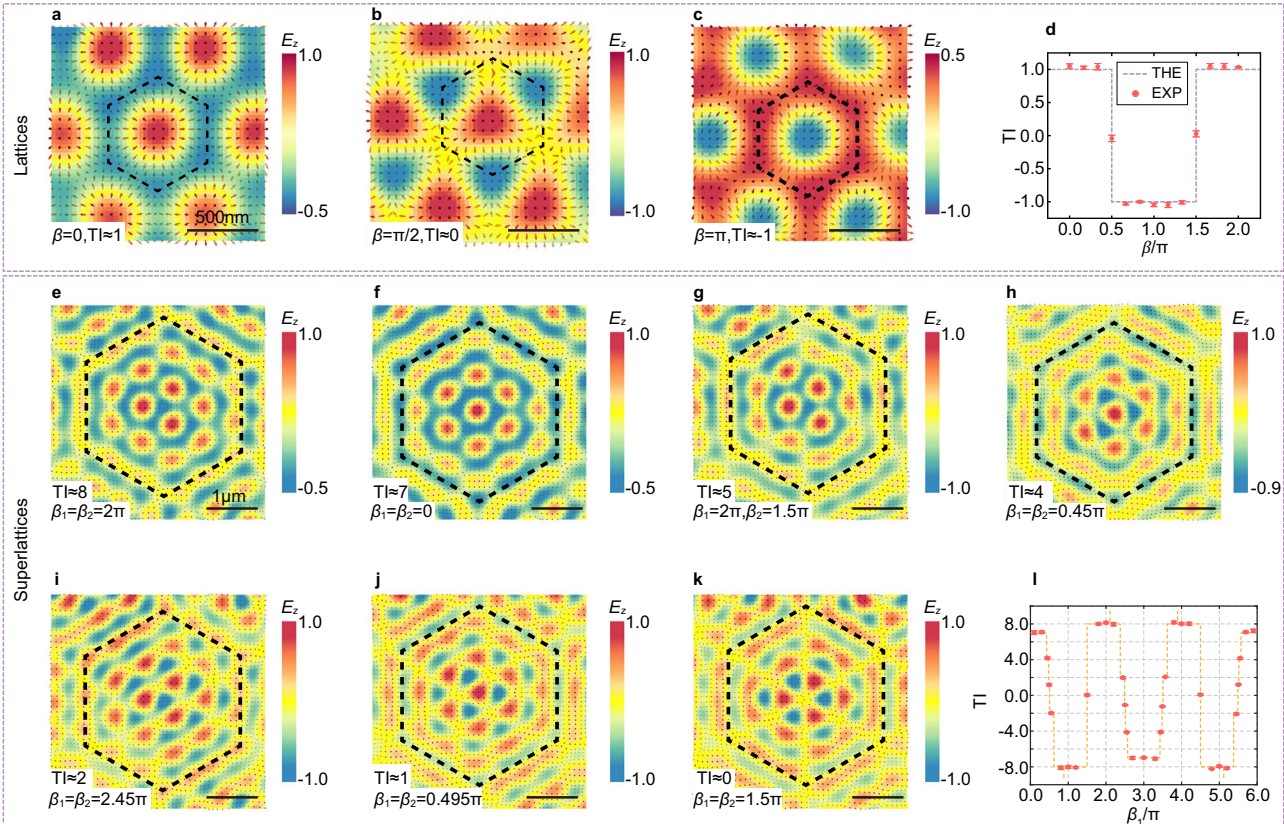

**Fig. 4 | Experimental demonstration of scalable and programmable topological transitions.** Experimenta $E_z$-field distributions described by Eq. (2), along with the reconstructed vector field patterns for $\beta$ values of 0 (**a**), $\pi/2$ (**b**), and $\pi$ (**c**), respectively. The unit cell is outlined by black dashed lines. **d** measured topological invariants (TIs) as a function of $\beta$ (dots), illustrating topological transition at $\beta = \pi/2$ and $3\pi/2$. The dashed line indicates theoretical results. Experimental $E_z$-field distributions described by Eq. (3), along with the reconstructed vector field patterns for TIs of 8 (**e**), 7 (**f**), 5 (**g**), 4 (**h**), 2 (**i**), 1 (**j**) and 0 (**k**), respectively. The unit cell is outlined by black dashed lines. **l** summary of the experimental values of topological invariants (discrete dots) when $\beta_1$ and $\beta_2$ are equal, illustrating that the TIs change several times in the range −8 to 8. The dashed line indicates theoretical results.

effect originates from the antisymmetry of vector field response to phase parameters. In contrast, the generic-position singularities can only modify the TI value in discrete steps of 3 (e.g., from −8 to −5, as shown with the dashed arrow in Fig. 2e). This behavior emerges from the interplay between the winding numbers of singularities and the $C_3$ symmetry of vector field (Supplementary Note 6).

These two mechanisms determine the permitted TT in the system. Based on this, we classify the system into two regimes: steady states and transient states. The steady states are regions of the parameter space ($\beta_1$, $\beta_2$) where no singularities are present, and the TI remains constant. The transient states are critical configurations where singularities emerge, and TTs occur between adjacent steady states. This helps to establish the fundamental constraint within the system, that is all steady-state TIs are forbidden from taking values that are integer multiples of 3. This restriction is followed from two observations. First, the extremal values of the TI spectrum, such as ± 8 in Fig. 2e, are not divisible by 3, and serve as boundary values for all reachable steady states. Second, the only allowed TI transitions—sign reversal and integer multiples of 3 changes—preserve the non-divisibility of TI by 3 throughout the entire parameter evolution. Therefore, the steady-state TI values form a discrete subset of the integers that exclude all multiples of 3. Furthermore, since transient states inherit their TI values as the arithmetic mean of adjacent steady states, they cannot take values that are integer multiples of 3/2 as well. As a result, the system exhibits a quantization rule that forbids integer multiples of 3/2 TI values, enforced by the topology and symmetry.

This exclusion rule is not restricted to a specific Moiré lattice configuration. It remains valid in a variety of systems with different ($m$, $n$) parameters. For instance, in configurations with ($m$, $n$)=(3, 4), the TI spans from −19 to +19, and in ($m$, $n$)=(4,9), from −58 to +58. In both cases, neither the steady-state TIs nor their averages are divisible by 3/2, thereby confirming the universality of this topological selection rule across system scales and symmetry classes. Note that the pattern of topological states is governed by the structural symmetry, as demonstrated by our investigation of multilayer Moiré superlattices. In multilayer hexagonal superlattices (Supplementary Note 7), the preservation of $C_3$ symmetry maintains a consistent pattern, even in complex multi-layer configurations. Conversely, the distinct symmetry of the square superlattices (Supplementary Note 8) yields a fundamentally different pattern of topological invariants.

**Experimental demonstration of scalable and programmable TTs**

We experimentally demonstrate scalable TTs using surface plasmon polaritons (SPPs)−TM-polarized evanescent waves propagating along a metal-dielectric interface[46]. The phase of each elementary SPP wave is dynamically controlled by a spatial light modulator, enabling the programmable construction of vector field topologies through control phase parameters. These code-like phase parameters function as effective degrees of freedom that map directly onto distinct topological states, thereby allowing flexible access to a wide range of TT. Detailed descriptions of the experimental setup and sample fabrication are provided in Methods.

The measured results for vector fields based on a hexagonal lattice configuration, governed by Eq. (2), are presented in Fig. 4a−c. The corresponding $E_z$-field distributions and reconstructed vector directions, obtained using the Gerchberg−Saxton (GS) algorithm[47], are

shown for three representative values of the encoded parameter: $\beta = 0$ (Fig. 4a), $\pi/2$ (Fig. 4b), and $\pi$ (Fig. 4c). The GS algorithm is a phase retrieval technique that has been used to reconstruct SPP fields with topological characteristics[48]. Additional results are summarized in Supplementary Figs. 3 and 4. Figure 4d depicts the TIs as a function of $\beta \in [0, 2\pi]$, with the dashed line indicating the theoretical predictions and the discrete points with standard deviations for experimental data. Clear TTs are observed at $\beta = \pi/2$ and $3\pi/2$, demonstrating programmable switching between distinct topological phases.

We proceed to analyze the experimental results of a more complex and higher-order topological system, the plasmonic Moiré superlattice for $(m, n) = (2, 3)$. Here, the topological state is jointly determined by two phase parameters, $\beta_1$ and $\beta_2$. As shown in Fig. 4e–k, this programmable approach allows continuous tuning of the system through a sequence of topological states with TIs of 8 (Fig. 4e), 7 (Fig. 4f), 5 (Fig. 4g), 4 (Fig. 4h), 2 (Fig. 4i), 1 (Fig. 4j), and 0 (Fig. 4k), respectively. Other cases are summarized in Supplementary Figs. 5 and 6. The extracted TIs (Fig. 4l), obtained for $\beta_1 = \beta_2$ swept from 0 to $6\pi$, illustrate the programmable and scalable evolution of multiple TTs within a single modulation cycle.

It is worth noting that the topological states of skyrmion-like vector fields are identified solely by their TIs, not by their visual resemblance. In Fig. 4, despite apparent morphological similarities among some field distributions, their differing TIs confirm that they represent distinct topological states. Furthermore, while losses are inevitable in the plasmonic platform, the topological structures remain robust on the wavelength scale, and topological transitions are clearly realizable due to the sufficiently low loss levels; a detailed analysis is provided in Supplementary Note 9.

## Discussion

We report large-range and programmable topological transitions (TTs) between distinct topological states of electromagnetic vector fields, enabled by Moiré-engineered optical skyrmion lattices. This work establishes a new platform for real-space topological control, featuring the broadest continuous tunability of TIs observed to date across all TT studies. Our results enrich the known forms of topological states and offer a clear physical picture of their dynamic evolution governed by singularity transformations. The universality of Moiré superlattices suggests that this framework can be extended beyond optics to other wave-based systems, including acoustics, photonics, gravitational waves, and quantum entanglement. In addition, the commonality between skyrmions and band structures provides the foundation for the construction of a general theoretical system of TTs, which opens new directions for developing next-generation electronic and photonic devices with enhanced compactness and performance.

## Methods
### Experimental configurations
The experimental setup used to probe the topological structures formed by SPP at the gold/air interface is shown in Supplementary Fig. 7. An incident laser beam ($\lambda = 632.8$ nm, Thorlabs, HNL050LB) was directed through a series of optical elements to produce six or twelve excitation beams with specific phases. Among them, a spatial light modulator (Meadowlark Optics E19x12-500-1200-HDM8, 1920 × 1200, pixel size: 8 × 8 µm) is used to generate six or twelve linearly polarized lights with specific phases. A half-wave plate and a vortex wave plate are used to generate the input radially polarized lights. These beams were tightly focused onto the sample, which consisted of a three-layer structure with a 50 nm-thick gold film on a silica substrate and the air as the upper medium. This focusing was achieved using an oil-immersion objective (Olympus, 100×, NA = 1.49), which excited the SPPs at the air/gold interface.

To map the local light field structure, silver nanoparticles with a diameter of 80 nm were immobilized on the gold surface. To spatially separate the weak SPP scattering from the nanoparticles from the directly transmitted background light, a circular opaque mask was inserted in the beam path to form a circular beam that illuminated the sample through a focusing objective. The intensity of the scattering radiation from the nanoparticle was measured using a photomultiplier tube (Hamamatsu, C12597-01), while the sample itself was mounted on a piezoelectric scanning stage (Physik Instrumente, P-727) with a resolution of 1 nm. The SPP field structure was obtained by scanning the nanoparticle across the beam profile. Due to the localized resonance properties of the silver nanoparticle-on-film gap structure, this probe reliably measures the out-of-plane electric field component of SPP[49].

### Sample preparation and data processing
The sample preparation followed these steps: first, a 5-nm-thick chromium layer was deposited onto glass coverslips (170 µm) by electron beam evaporation to improve adhesion. Subsequently, a 50-nm-thick gold film was deposited using the same method. Finally, a droplet of a fully diluted suspension of silver nanoparticles (80 nm in diameter) was placed on the surface of the gold film and allowed to evaporate naturally. While the dried sample contained numerous silver nanoparticles, measurements were exclusively performed on individual particles isolated by more than 20 µm from neighbors, ensuring that all scattered light originated from a single target nanoparticle. No additional fixation is required, as natural drying provides enough stability for measurement. The data was acquired at a sampling density of either 10 nm or 30 nm. This choice of sampling density is determined by the spatial variation rate of the vector field and experimental conditions. Theoretically, a higher density of data points provides a more accurate description of the vector field. However, denser data sampling is more susceptible to environmental influences. Even slight vibrations can introduce unintended frequency artifacts into the experimental data. Therefore, for rapidly varying fields, we typically use a 10 nm resolution, whereas a 30 nm resolution is generally sufficient for most common cases.

The original mapped intensity distributions of the out-of-plane electric field $|E_z|^2$, excited on the surface of the gold film under six-beam and twelve-beam excitation light, are presented in Supplementary Figs. 3 and 5, respectively. After acquiring the experimental data, the real part of $E_z$ and the corresponding phase patterns were reconstructed using the Gerchberg-Saxton (GS) algorithm[47,48], as shown in Supplementary Figs. 4 and 6. Subsequently, the $x$-component $E_x$ and $y$-component $E_y$ were calculated according to Eq. (S4) of the supplementary information. The TIs of the vector fields were then computed using Eq. (1) from the main text, and the discrete results are shown in Fig. 4d, l of the main text.

## Data availability
Source data for all figures are provided with this paper (ref. 50).

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

## Acknowledgements

This work was supported by Science and Technology Innovation Commission of Shenzhen (Grant No. RCJC20200714114435063, to L.D.); Science and Technology Innovation Commission of Shenzhen (Grant No. JCYJ20241202124532024; to L.D.); Innovation Team Project of Ordinary University of Guangdong Provincial Education Bureau (Grant No. 2024KCXTD014; to L.D.); Research Team Cultivation Program of Shenzhen University (Grant No. 2023QNT012; to L.D.); Shenzhen Youth Talent Foundation (Grant No. RCYX20231211090249068; to X.Z.); National Natural Science Foundation of China (Grant No. 52275565; to X.Z.); National Natural Science Foundation of China (Grant No. 12404347; to R.W.); Guangdong Major Project of Basic Research (Grant No. 2020B0301030009; to X.Y.); Shenzhen University 2035 Initiative (Grant No. 2023B004; to X.Y.)

## Author contributions

B.T. and L.D. developed the concept of the work. B.T. and L.D. designed the experiment. B.T. performed the measurements. B.T., X.Z. and L.D. built the theoretical model and performed the analytical calculations. B.T., X.Z., R.W., Y.Z., X.Y., and L.D. wrote the manuscript. All the authors discussed the results and commented on the manuscript.

## Competing interests

The authors declare no competing interests.
