## [Peer Review file · Nature Communications]

Scalable and Programmable Topological Transitions in Plasmonic Moiré Superlattices

Corresponding Author: Professor Luping Du

Version 0:

Reviewer comments:

Reviewer #1

(Remarks to the Author)

The authors report on an experimental and theoretical framework for achieving scalable and programmable topological phase transitions (TPTs) in plasmonic systems. The study utilizes the interference of six or twelve evanescent waves to form skyrmion vector fields. By manipulating the phases of the individual waves, the authors achieve tunable topological invariants (TIs). The selection rule of possible TI's has been derived, revealing that changing the synthetic phase parameter in this system only allows for (integer and half-integer) TI's that exclude integer multiples of $3/2$. The work also aims to bridge momentum-space topological band theory with real-space skyrmion topologies.

The focus of this work on topological phase transitions in plasmonic Moiré superlattices sits at the intersection of several advanced fields—nanophotonics, topological physics, and solid-state physics. That said, for specialized communities—especially those working on topological photonics, metasurfaces, or plasmonics—the paper is both interesting and significant.

The authors are well-known for their expertise in nanophotonics research and present high-quality experimental results. The manuscript is well written, and the data presented well. Due to its novelty, we believe the manuscript could be suitable for Nature Communications.

Nevertheless, before making a final decision, we believe that certain points need to be addressed:

Main concerns:

Not only the concept of plasmonic skyrmion moiré superlattices, but also the fact that they enable arbitrarily large TIs in their super unit cells has recently been introduced in Refs. 37, 38.

However, the ability to tune these TI's continuously using a wisely defined synthetic phase parameter is novel. In our opinion, the authors could emphasize the similarities and differences more clearly to set themselves apart from these previous works.

We suggest broadening the study to additional configurations of moiré superlattices that have been introduced. In particular, different centers of rotation could change the range of accessible TIs and their selection rules. The authors could also discuss the effect of using three or more twisted lattices and the resulting influence on the TIs and selection rules.

The authors claim in the introduction that 'the connection between the TPTs of skyrmions and the TPTs of energy bands has also been identified' and in the conclusion that 'the commonality between skyrmions and band structures provides the foundation for the construction of general theoretical system of TPTs, which opens new directions for developing next-generation electronic and photonic devices with enhanced compactness and performance.'

Unfortunately, in the main text only little attention has been drawn to this connection and its implications.

It is commonly known that singularities in the field (i.e., $|E|=0$) are a necessary condition for TPTs. In the SI, the authors now

introduce energy band-like structures, that follow $\pm\sqrt{(E_x^2+E_y^2+E_z^2)} = \pm|E|$. From this definition of the bands, it is a straightforward conclusion that TPTs are connected to a gap-free structure as they require $|E|=0$.

Can the authors state if such energy band-like structures have already been defined in other works? We think that the definition of these energy band-like structures should be included in a few sentences of the main text, as the authors highlight this finding as one of the main novelties of their work.

In addition, their connection to field singularities is straightforward and should not be hidden in the supplementary material.

How universal can these energy band-like structures be applied? Are there any applications and use cases? Is it, for example, possible to 'simulate' a two-band model using a plasmonic system and connect any parameters of such a solid-state system to the topology of the plasmonic system?

The authors also did not include such energy band-like structures for the moiré systems. Are there any conclusions that can be drawn from the topology of these bands, and are there any connections to moiré materials?

Apart from these points, we have some additional comments and questions:

In our opinion, it is necessary that the authors present the calculated Chern number density to support the calculated Chern numbers (i.e., TIs) from their analytical and experimental data. We suggest that the authors include these figures in the supplementary material. The field singularities should be connected to singularities and a change of the sign in the local Chern number density. The authors should also include a short explanation of how the Chern numbers are calculated from the Chern number density. Are they calculated by summing up all measured pixels of the Chern number density inside the super unit cell boundary, or are there any additional numerical methods involved?

The authors provide experimental confirmation of programmable TPTs using a nanoparticle-scattering technique, retrieving the amplitude of the out-of-plane field (E_z). The full vector fields are reconstructed using the Gerchberg-Saxton algorithm and Maxwell's equations.

In our opinion, the authors could provide some more information about their experimental methods.

What is the resolution of the measured field distribution? What are its limitations?

How many silver nanoparticles are located on the gold surface, and how are they immobilized? How many of these nanoparticles are involved in the measurement?

The GS algorithm is primarily applied to retrieve the phase from intensity measurements that are acquired in two different planes. Typically, the image plane and the far field plane. Can the authors provide more information on the usage of the GS algorithm to retrieve the phase of SPPs, or reference another work where this is explained? Does the GS algorithm always converge to the desired solution?

The authors state in line 120: 'This relation reveals a periodicity of 2π , within which three distinct topological states emerge, characterized by TIs of 1, 0, and -1 . Specifically, the TI remains unchanged as β varies within several open intervals, such as $(-\pi/2, \pi/2)$ and $(\pi/2, 3\pi/2)$, indicating the robustness of the topological state.'

Could the authors elaborate on the robustness that is given by the half-open interval? Does this mean that the TI of 0 is more difficult to create or measure?

Minor concerns:

Line 81: remove 'distint'.

Line 90 and Supplementary Section 2: 'warps' should be replaced by 'wraps'.

Supplementary Section 2: 'Soling' should be replaced by 'Solving'.

Line 167: replace 'simultaneous' with 'simultaneously'.

Line 216: 'This scaling behavior underscores the capacity of Moiré structuring to support a wide spectrum of topological states (Supplementary Information Section 5).'

This sentence should be rephrased to e.g:

'This scaling behavior highlights the ability of Moiré structuring to support a broad range of topological states

(Supplementary Information, Section 5).'

Line 285, 289: 'The unit cell is outlined...'

Line 294: remove 'nest'.

(Remarks to the Author)
my comments are given in the attached pdf file.

Version 1:

Reviewer comments:

Reviewer #1

(Remarks to the Author)
The authors have improved the manuscript accordingly and it can now be accepted for publication.

Reviewer #2

(Remarks to the Author)
I read with great pleasure the response of the authors to my comments and to the comments of the other reviewer. I believe they have adequately addressed our concerns, and I recommend that their manuscript be published in Nature Communications.

Reviewer #3

(Remarks to the Author)
I co-reviewed this manuscript with one of the reviewers who provided the listed reports. This is part of the Nature Communications initiative to facilitate training in peer review and to provide appropriate recognition for Early Career Researchers who co-review manuscripts.

In cases where reviewers are anonymous, credit should be given to 'Anonymous Referee' and the source. The images or other third party material in this Peer Review File are included in the article's Creative Commons license, unless indicated otherwise in a credit line to the material. If material is not included in the article's Creative Commons license and your intended use is not permitted by statutory regulation or exceeds the permitted use, you will need to obtain permission directly from the copyright holder.

Dear Editors and Reviewers:

We thank the reviewers for their time and comments regarding our manuscript entitled "*Scalable and Programmable Topological Phase Transitions in Plasmonic Moiré Superlattices*" (ID: NCOMMS-25-44015). We appreciate their positive feedback and have found the suggestions to be inspiring and helpful in improving the quality of our manuscript. We have thoroughly looked through every comment and have revised the manuscript accordingly. We hope our revisions have improved our manuscript to guarantee its publication in *Nature Communications*. All modifications have been highlighted in the text, followed by a detailed point-by-point response for each comment.

Response to Reviewer #1:

Comment: The authors report on an experimental and theoretical framework for achieving scalable and programmable topological phase transitions (TPTs) in plasmonic systems. The study utilizes the interference of six or twelve evanescent waves to form skyrmion vector fields. By manipulating the phases of the individual waves, the authors achieve tunable topological invariants (TIs). The selection rule of possible TI's has been derived, revealing that changing the synthetic phase parameter in this system only allows for (integer and half-integer) TI's that exclude integer multiples of $3/2$. The work also aims to bridge momentum-space topological band theory with real-space skyrmion topologies.

The focus of this work on topological phase transitions in plasmonic Moiré superlattices sits at the intersection of several advanced fields—nanophotonics, topological physics, and solid-state physics. That said, for specialized communities—especially those working on topological photonics, metasurfaces, or plasmonics—the paper is both interesting and significant.

The authors are well-known for their expertise in nanophotonics research and present high-quality experimental results. The manuscript is well written, and the data presented well. Due to its novelty, we believe the manuscript could be suitable for *Nature Communications*.

Nevertheless, before making a final decision, we believe that certain points need to be addressed:

Q1: Not only the concept of plasmonic skyrmion Moiré superlattices, but also the fact that they enable arbitrarily large TIs in their super unit cells has recently been introduced in Refs. 37, 38.

However, the ability to tune these TI's continuously using a wisely defined synthetic phase parameter is novel. In our opinion, the authors could emphasize the similarities

and differences more clearly to set themselves apart from these previous works. We suggest broadening the study to additional configurations of Moiré superlattices that have been introduced. In particular, different centers of rotation could change the range of accessible TIs and their selection rules. The authors could also discuss the effect of using three or more twisted lattices and the resulting influence on the TIs and selection rules.

R1: Thank you for the constructive suggestion. We are sorry for the oversight in not emphasizing the importance of the cited Refs. 37 and 38, as both of them are outstanding works related to the development of plasmonic skyrmion Moiré superlattices. We certainly agree with the reviewer that clarifying the relationship between our work and the Refs. 37, 38 is crucial for highlighting the novel contribution of our study. We have now emphasized the impact of these two Refs in the introduction section when discussing the plasmonic Moiré superlattice. The specific modifications made are as follows:

In particular, the Moiré superlattices generated in plasmonic platforms^{40,41} can support arbitrarily large topological invariants. This capability not only significantly enriches the diversity of topological states but also facilitates topological transitions between these states.

The detailed content can be found in lines 52 to 55 of the main text (Refs. 37 and 38 in the initial version are now Refs. 40 and 41, respectively, due to structural adjustments).

Furthermore, we sincerely thank the reviewer for the suggestion of extending the study to additional configurations, as this will certainly provide a deep insight into the relationship between topological states and symmetry in such a system. Therefore, in the Supplementary Information Sections 7 and 8, we present topological states in multilayer hexagonal Moiré superlattices and square Moiré superlattices, respectively. The results demonstrate that in multilayer hexagonal Moiré superlattices, even with complex structures, the pattern governing the existence of topological invariants remains unchanged due to the preservation of C_3 symmetry. In contrast, the distinct symmetry of square Moiré superlattices leads to a different pattern of topological states, where phase variations lead to a zero value of the topological invariant. This phenomenon arises because the singularities generated in the square Moiré superlattice are created in pairs with opposite winding numbers. Consequently, these singularities cancel each other out, leading to no net change in the topological state. To further clarify that the patterns of topological invariants in our work are specific to certain symmetries, we have emphasized this point in the main text. The specific modifications made are as follows:

Note that the pattern of topological states is governed by the structural symmetry, as demonstrated by our investigation of multilayer Moiré superlattices. In multilayer hexagonal superlattices (Supplementary Information Section 7), the preservation of C_3

symmetry maintains a consistent pattern, even in complex multi-layer configurations. Conversely, the distinct symmetry of the square superlattices (Supplementary Information Section 8) yields a fundamentally different pattern of topological invariants.

The detailed content can be found in lines 294 to 300 of the main text.

Q2: The authors claim in the introduction that ‘the connection between the TPTs of skyrmions and the TPTs of energy bands has also been identified’ and in the conclusion that ‘the commonality between skyrmions and band structures provides the foundation for the construction of general theoretical system of TPTs, which opens new directions for developing next-generation electronic and photonic devices with enhanced compactness and performance.’

Unfortunately, in the main text only little attention has been drawn to this connection and its implications.

It is commonly known that singularities in the field (i.e., $|E|=0$) are a necessary condition for TPTs. In the SI, the authors now introduce energy band-like structures, that follow $\pm\sqrt{E_x^2 + E_y^2 + E_z^2} = \pm|E|$. From this definition of the bands, it is a straightforward conclusion that TPTs are connected to a gap-free structure as they require $|E|=0$.

Can the authors state if such energy band-like structures have already been defined in other works? We think that the definition of these energy band-like structures should be included in a few sentences of the main text, as the authors highlight this finding as one of the main novelties of their work.

In addition, their connection to field singularities is straightforward and should not be hidden in the supplementary material.

How universal can these energy band-like structures be applied? Are there any applications and use cases? Is it, for example, possible to ‘simulate’ a two-band model using a plasmonic system and connect any parameters of such a solid-state system to the topology of the plasmonic system?

R2: We sincerely thank the reviewer for the insightful suggestions. We apologize for not elaborating on the connection between our work on these aspects previously. We have now incorporated a discussion and analysis of the energy band-like structure into the main text to emphasize the significance of this feature in topological transitions. The specific modifications made are as follows:

The mechanism underlying the TT of optical skyrmions mirrors that underlying

topological phase transitions in topological insulators—both are driven by the emergence of singularities in the system. This commonality is elucidated by analyzing the topological invariants of skyrmions and comparing them to those arising in the two-band model used for topological insulators. Within topological insulators, the two-band model, which captures the evolution of the band structure across topological phase transitions, features energy bands of the form $\pm|\mathbf{h}|$. The Chern number characterizing the topological nature of the band structure is given by a formula identical to that in Eq. (1). Here, \mathbf{h} denotes a vector in the Brillouin zone parameterized by the hopping strengths. Consequently, the Chern number can be interpreted as the topological invariant of a skyrmion by treating \mathbf{h} as a mapping from the first Brillouin zone to the real-space sphere S^2 . In topological insulators, a topological phase transition is triggered when the energy bands close and reopen. Band closure implies that the vector \mathbf{h} must vanish ($\mathbf{h}=0$) at one or more points in the Brillouin zone. By extending this concept to vector fields defined on a hexagonal lattice, we deduce that a necessary condition for a TT in such a system is similarly the emergence of singularities within the vector field. A detailed theoretical treatment is provided in Supplementary Information Section 2.

The detailed content can be found in lines 138 to 154 of the main text.

Specifically, to the best of our knowledge, such an energy band-like structure has not been explicitly reported in the field of optics. We introduce this concept based on the analogy between the skyrmion number and the Chern number in the two-band model. According to the topological phase transition mechanism of the two-band model, this band-like structure provides a generally applicable framework for describing changes in the topological states of vector fields in lattice systems. In other words, irrespective of the specific lattice configuration, this band-like structure can serve as a useful means to characterize the process of topological transition.

However, it should be emphasized that this band-like structure shares only mathematical similarities with the energy band structure of the two-band model. In this mathematical analogy, the conduction band in the energy band structure corresponds to the magnitude of the vector field $|\mathbf{E}|$ in the band-like structure, while the valence band corresponds to the symmetric distribution of $|\mathbf{E}|$ with respect to the $z=0$ plane. Likewise, the momentum space (k_x, k_y, k_z) in the energy band model maps onto the real space (x, y, z) in the band-like structure.

The underlying physical implications between band-like structure and band structure in topological insulators are fundamentally distinct. Therefore, to date, we have not identified a viable approach to utilize plasmonic systems for simulating the physical phenomena described by the two-band model. Nevertheless, we consider the reviewer's suggestion—to employ plasmonic systems in simulating the two-band model—a highly valuable insight. We hypothesize that the mathematical analogy revealed in our manuscript may lay the groundwork for unifying two distinct types of topological

structures within a more generalized framework. Such unification could, in turn, potentially enable mutual simulation between these two topological systems.

Q3: The authors also did not include such energy band-like structures for the Moiré systems. Are there any conclusions that can be drawn from the topology of these bands, and are there any connections to Moiré materials?

R3: We thank the reviewer for the keen observation and helpful suggestion. In the revised Supplementary Fig. 2, we have included the energy band-like structures associated with the hexagonal Moiré superlattices. Moreover, for the newly analyzed three-layer hexagonal and square Moiré superlattices, the band-like structures have also been incorporated and illustrated in Supplementary Figs. 11 and 12, respectively.

Analysis of these frameworks, in both hexagonal lattices and hexagonal Moiré superlattices, reveals a fundamental principle of topological transitions: during the closing and reopening of the energy band-like structure, both the number and the nature of the singularities collectively determine the topological invariant of the vector field after the topological transition. Consequently, a greater number of singularities of the same type leads to a more significant change of the topological invariant. Given that the number of singularities in hexagonal Moiré superlattices is significantly higher than that in hexagonal lattices, they enable a broader range of variation in topological invariants.

The Moiré superlattices discussed in this work and Moiré metamaterials share a conceptual connection at the research level, even though their specific physical mechanisms and application scenarios differ from each other. Nevertheless, we note that these two fields can mutually inspire and inform each other. For example, the discovery of unconventional superconductivity in Moiré graphene sparked tremendous interest, subsequently fueling extensive research into both Moiré metamaterials and Moiré superlattices. Many later studies drew inspiration, to varying extents, from the fundamental ideas established in Moiré graphene. Similarly, the observation of diverse topological transitions and multiple topological states in optical hexagonal Moiré superlattices suggests that this conceptual framework could be extended to other Moiré systems and metamaterials. In particular, mapping the phase modulation strategies from Moiré superlattices onto Moiré metamaterials may enable the realization of rich topological states in such systems, paving the way for the development of topological materials with enhanced or novel properties.

Q4: In our opinion, it is necessary that the authors present the calculated Chern number density to support the calculated Chern numbers (i.e., TIs) from their analytical and experimental data. We suggest that the authors include these figures in the supplementary material. The field singularities should be connected to singularities and a change of the sign in the local Chern number density. The authors should also include a short explanation of how the Chern numbers are calculated from the Chern number

density. Are they calculated by summing up all measured pixels of the Chern number density inside the super unit cell boundary, or are there any additional numerical methods involved?

R4: Another great suggestion from the reviewer. We have now included the topological charge density in Supplementary Fig. 4 and 6.

Here, Supplementary Fig. R1 is attached to illustrate the topological invariants of skyrmions from experimental calculation. We note that Supplementary Fig. R1 presents the data presented in the last column of Supplementary Fig. 4, namely, the topological charge density of skyrmions in hexagonal lattices. The topological invariant is computed through a discrete approximation of the surface integral: the topological charge density is summed over all points within a unit cell, multiplied by the area element, and normalized by 4π . This operation represents a direct discretization of the continuous integral and involves no additional processing steps. To more clearly and conveniently illustrate the details of experimental data processing, we have additionally described the method for calculating topological invariants using experimental data in the caption of Supplementary Figs. 4 and 6.

Supplementary Fig. R1| Topological charge density of skyrmions in hexagonal lattices.

Supplementary Fig. R1 also reveals that even in the simplest hexagonal lattice ($\beta=0$), the topological charge density exhibits a nonuniform distribution, taking both positive and negative values. The negative values are localized near the six vertices of the unit cell. As β increases, the regions of negative topological charge density around three non-adjacent vertices expand, while those around the other three vertices gradually shrink. When β reaches $\pi/2$, the negative-charge regions associated with three non-adjacent vertices extend to occupy half of the unit cell, whereas the other three vanish entirely. The singularities are located at the three non-adjacent vertices where the negative topological charge vanishes. Because the six vertices are shared by three adjacent unit cells, the unit cell under observation effectively contains only one singularity, and the negative-charge regions occupy exactly 50% of the cell area. This complete balance between positive and negative contributions yields a topological invariant of zero.

As β increases beyond $\pi/2$ and approaches π , the evolution of the topological charge density mirrors the behavior observed from $\beta=0$ to $\pi/2$, but with opposite sign. In other

words, the evolution of the negative (positive) charge density regions for β increasing from 0 to $\pi/2$ corresponds precisely to that of the positive (negative) regions as β increases from π down to $\pi/2$.

In more complex Moiré superlattices (Supplementary Fig. 6), the distribution of topological charge density becomes more intricate, yet its evolution follows the same general principle. Each topological state features regions of both positive and negative topological charge density, which evolve continuously with variations in the phase parameters. The topological state of the system is jointly determined by the interplay between these regions. When a singularity emerges during the phase evolution, the collective contribution of the charge density undergoes an abrupt change, thereby triggering a transition in the topological state of the vector field.

Q5: The authors provide experimental confirmation of programmable TPTs using a nanoparticle-scattering technique, retrieving the amplitude of the out-of-plane field (E_z). The full vector fields are reconstructed using the Gerchberg-Saxton algorithm and Maxwell's equations. In our opinion, the authors could provide some more information about their experimental methods.

What is the resolution of the measured field distribution? What are its limitations?

How many silver nanoparticles are located on the gold surface, and how are they immobilized? How many of these nanoparticles are involved in the measurement?

The GS algorithm is primarily applied to retrieve the phase from intensity measurements that are acquired in two different planes. Typically, the image plane and the far field plane. Can the authors provide more information on the usage of the GS algorithm to retrieve the phase of SPPs, or reference another work where this is explained? Does the GS algorithm always converge to the desired solution?

R5: We thank the reviewer for the interest in our experimental details, and we would like to take this opportunity to include necessary clarifications. We have expanded the section detailing the experimental procedures in the main text. The specific modifications made are as follows:

The sample preparation followed these steps: first, a 5-nm-thick chromium layer was deposited onto glass coverslips (170 μm) by electron beam evaporation to improve adhesion. Subsequently, a 50-nm-thick gold film was deposited using the same method. Finally, a droplet of a fully diluted suspension of silver nanoparticles (80 nm in diameter) was placed on the surface of the gold film and allowed to evaporate naturally. While the dried sample contained numerous silver nanoparticles, measurements were exclusively performed on individual particles isolated by more than 20 μm from neighbors, ensuring that all scattered light originated from a single target nanoparticle. No additional fixation is required, as natural drying provides enough stability for

measurement. The data was acquired at a sampling density of either 10 nm or 30 nm. This choice of sampling density is determined by the spatial variation rate of the vector field and experimental conditions. Theoretically, a higher density of data points provides a more accurate description of the vector field. However, denser data sampling is more susceptible to environmental influences. Even slight vibrations can introduce unintended frequency artifacts into the experimental data. Therefore, for rapidly varying fields, we typically use a 10 nm resolution, whereas a 30 nm resolution is generally sufficient for most common cases.

The detailed content can be found in lines 385 to 401 of the main text.

In the phase retrieval process using the Gerchberg-Saxton algorithm, the initial phase guess is set to zero for all pixels. According to Eqs. (S3) and (S18), the phase retrieved from the measured intensity data can only take on values of 0 and $\pm\pi$. Therefore, this default phase guess also aids the algorithm in converging to the correct solution. Additionally, our experience shows that a more uniform intensity distribution leads to better phase retrieval performance. Thus, during experimental data acquisition, calibrating the optical path and selecting stable particles with high responsiveness are crucial for obtaining a recorded intensity distribution that is as uniform as possible. Since the structures proposed in this manuscript, whether hexagonal lattices or hexagonal Moiré superlattices, the vector field at any point exhibits continuously distributed function values and derivatives. Consequently, the algorithm converges to the target value when the experimental data exhibits no significant intensity distribution imbalance.

We also note that the Gerchberg-Saxton algorithm has been widely employed to reconstruct the phase in surface plasmon polaritons, for example, *Nature Photonics*, 15, 442–448 (2021). The foundational principles of the algorithm are detailed in *Optik* 34, 3 (1971); *Optik* 35, 2 (1972); and *Optics Letters* 3, 1 (1978).

Q6: The authors state in line 120: ‘This relation reveals a periodicity of 2π , within which three distinct topological states emerge, characterized by TIs of 1, 0, and -1 . Specifically, the TI remains unchanged as β varies within several open intervals, such as $(-\pi/2, \pi/2)$ and $(\pi/2, 3\pi/2)$, indicating the robustness of the topological state.’ Could the authors elaborate on the robustness that is given by the half-open interval? Does this mean that the TI of 0 is more difficult to create or measure?

R6: We thank the reviewer for raising this and we would take the opportunity to clarify this. Robustness refers to the property whereby a system’s topology remains invariant under perturbations. For the skyrmions formed in the hexagonal lattice proposed in this work, we find that the topological state of the vector field remains unchanged across the phase parameter intervals $(-\pi/2, \pi/2)$ or $(\pi/2, 3\pi/2)$. The persistence of the topological state under continuous variations of β within these ranges clearly demonstrates its robustness against phase perturbations.

In the hexagonal lattice, the topological state characterized by a $TI = 0$ represents a critical point between two distinct topological phases, $TI = +1$ and $TI = -1$, occurring specifically at $\beta = \pi/2$. Theoretically, each state can be uniquely defined by a specific interval or discrete value of β . However, in practical implementations, the topological states with $TI = \pm 1$ exhibit higher tolerance to experimental imperfections than the critical state. This arises because, in real systems, the implemented phase is often influenced by external factors, such as inhomogeneities in the refractive index of the medium. When such perturbations induce small deviations in phase parameters, the $TI = \pm 1$ states—corresponding to broad, stable phase intervals—remain unaffected by minor variations. In contrast, the critical $TI = 0$ state is inherently unstable: even slight perturbations can drive it to transition abruptly into one of the neighboring topological states.

Q7: Line 81: remove ‘distint’.

Line 90 and Supplementary Section 2: ‘warps’ should be replaced by ‘wraps’.

Supplementary Section 2: ‘Soling’ should be replaced by ‘Solving’.

Line 167: replace ‘simultaneous’ with ‘simultaneously’.

Line 216: ‘This scaling behavior underscores the capacity of Moiré structuring to support a wide spectrum of topological states (Supplementary Information Section 5).’

This sentence should be rephrased to e.g: ‘This scaling behavior highlights the ability of Moiré structuring to support a broad range of topological states (Supplementary Information, Section 5).’

Line 285, 289: ‘The unit cell is outlined...’.

Line 294: remove ‘nest’.

R7: Thank you very much for the careful and thoughtful review of our manuscript. We sincerely apologize for any unintentional errors and would like to confirm that we have thoroughly re-examined the entire manuscript and corrected all identified issues.

Response to Reviewer #2:

Comment: I read with great interest the manuscript by B. Tian et al, about implementation of high-order topological invariants in plasmonic fields using Moiré superlattices. Although the idea itself is not new (first performed in Ref. 37 of the manuscript), the authors extend the known theory and understanding of this phenomenon, and showcase control over the topological invariant via an SLM. They characterize the reason for transitions between topological numbers and demonstrate their theory through an experiment.

I find that the authors’ idea is innovative, interesting to a broad audience in photonics and topological physics, and – for the most part –performed rigorously (see my comments below for a few “loose-ends”). I am inclined to recommend publication in principle, given that the authors adequately address all comments below:

Q1: The authors call the transitions in their system “phase transitions”. It is not an accurate phrase, since it is not a many-body system, nor does it contain any form of nonlinearity. The cited works in topological photonics certainly do not show phase transitions, but rather – just “topological transitions”, which are the commonly agreed upon term. I request that the authors amend their manuscript in accordance with this point.

R1: Thank you for the insightful suggestion. Our initial use of the term "topological phase transition" was inspired by condensed-matter physics, where it describes transitions between distinct topological phases in electronic systems. This terminology was adopted because the underlying mathematical framework—particularly the analogy between the Chern number and the skyrmion number in the two-band model—is similar. However, we acknowledge that the physical mechanisms in our plasmonic Moiré system differ from those in condensed-matter systems. To ensure conceptual accuracy, we have therefore replaced "topological phase transition" with "topological transition" throughout the whole manuscript, including in the title.

Q2: The claim of showing topological invariants between -58 and 58 is unsubstantiated. The authors show this in theory, while measuring only -8 to 8 . I suggest that either the authors amend their claim, or properly measure topological invariants up to 58 .

R2: We are grateful to the reviewer for the valuable suggestion. We have revised the manuscript to clarify the distinction between theoretical predictions and experimental verification regarding the range of topological invariants. In theory, our calculations show that very large-scale topological transitions can be achieved by changing the Moiré angle. Experimentally, we verified the occurrence of these transitions at a specific Moiré angle (13.17°), confirming the predicted topological changes. The relevant section now reads as follows:

Theoretical calculations indicate that TIs occur between skyrmion configurations carrying integer and half-integer topological invariants (TIs) within the range of -8 to $+8$, excluding TIs that are integer multiples of $3/2$ at a Moiré angle of 13.17° . The accessible range of TIs further expands to -19 to $+19$ at 9.43° , and reaches -58 to $+58$ at 25.04° , representing one of the largest-scale TIs reported to date in physical systems. This selection rule, governed by the discrete symmetry of the lattice, reflects an intrinsic constraint on real-space topological quantization. Experimentally, we investigated the evolution of the topological states of the vector field in the Moiré superlattice at a Moiré angle of 13.17° . Although measurements were conducted at this single angle, the experimental results clearly confirm the occurrence of the predicted topological transitions, consistent with the theoretical calculations regarding the presence and distribution of these topological states.

The detailed content can be found in lines 62 to 73 of the main text.

Q3: I greatly appreciate that the authors check whether or not their minimal field amplitudes tend to 0. In such points, the topological density is ill-defined, so the whole distribution cannot have a well-defined topological charge other than 0. In this respect, any fractional topological charge the authors measure is probably occurring in such a point, and therefore – is ill-defined. In addition, is it true that the dashed lines in Fig. 2e coincide with zero minimal field amplitude lines shown in Fig. 2f ? this would mean that, along these lines, the topological constant is ill-defined, whereas right now it is defined via a particular color in Fig. 2e. Therefore, the figure may have to change, and the authors should complete their check that indeed the topological density in cases they measure is well-defined.

R3: We appreciate this particularly insightful comment from the reviewer. Our detailed analysis of the vector field variation patterns shows that its amplitude can indeed reach zero. To illustrate this, we consider a singularity within a hexagonal lattice. The vector field components in such a lattice are given by:

$$E_z = E_0 \sum_{\alpha=1}^3 \cos[k_s x \cos(\phi_\alpha) + k_s y \sin(\phi_\alpha) + \varphi_\alpha], \quad (\text{R1a})$$

$$E_x = E_0 \sum_{\alpha=1}^3 \frac{k_z}{k_s} \cos(\phi_\alpha) \sin[k_s x \cos(\phi_\alpha) + k_s y \sin(\phi_\alpha) + \varphi_\alpha], \quad (\text{R1b})$$

$$E_y = E_0 \sum_{\alpha=1}^3 \frac{k_z}{k_s} \sin(\phi_\alpha) \sin[k_s x \cos(\phi_\alpha) + k_s y \sin(\phi_\alpha) + \varphi_\alpha], \quad (\text{R1c})$$

where $\varphi_2 = \beta/3$, $\varphi_1 = \varphi_3 = -\beta/3$, and the phases ϕ_1, ϕ_2, ϕ_3 are successively separated by $\pi/3$. For the case of $\beta=\pi/2$, with $\phi_1 = \pi/6$, $\phi_2 = \pi/2$ and $\phi_3 = 5\pi/6$, all three components of the vector field in Eq. (R1) vanish simultaneously at the position ($k_s x=0, k_s y=4\pi/3$). This confirms that the vector field amplitude reaches a minimum of zero.

Despite the presence of these singularities, a well-defined topological invariant exists and effectively characterizes the system. As shown in Supplementary Fig. R2a for $\beta=\pi/2$, the vector field contains singularities. Length-normalized arrows indicate vector directions, with points "a," "b," and "c" marking three singularities located at the vertices of the lattice unit cell. To better illustrate the variation trends near singularities, Supplementary Fig. R2b zooms in on the region around singularity "a." At this singularity, all three vector components vanish, leaving the vector direction undefined at that point. This implies that across the unit cell, the topological charge density is undefined only at these isolated singularities, while remaining well-defined elsewhere.

Supplementary Fig. R2| Distribution of Vector Fields and Singularities in hexagonal Lattice. **a**, The electric field distribution for $\beta=\pi/2$. Length-normalized arrows indicate the vector directions, with points "a", "b", and "c" marking three singularities located at the vertices of the lattice unit cell. **b**, Zoomed-in view of the region around singularity "a".

The topological invariant for skyrmions is computed via a surface integral. Importantly, the presence of a finite number of singularities within the integration region does not affect the integral's outcome. Therefore, the overall structure can still be robustly characterized by a topological invariant, even in the presence of singularities.

Furthermore, our analysis shows that the dashed lines in Fig. 2e correspond precisely to the zero-value contours in Fig. 2f, indicating the presence of isolated singularities within the unit cell. As discussed above, these singularities do not influence the calculation of the topological invariant. In summary, for any value of the phase parameter, a well-defined topological invariant exists to characterize the topological state of the vector field.

Q4: In the same vein of checking that the distribution is well-defined, how do the authors make sure that the real part of the electric field is indeed much larger than the imaginary part? Of course, this can be defined up to a global phase, but to take only one part of the field, it must be much larger than the other part, with the criterion defined in the supplementary information of Ref. 22. It seems, from looking at the supplementary, that this might not be the case for some of the measurements. I strongly recommend that the authors check this point to make sure that their main claims are substantiated.

R4: We thank the reviewer for providing us with the opportunity to clarify this important point. As shown in Figs. 1a and 2b of the main text, the phase relationships in both the hexagonal lattice and the hexagonal Moiré lattice indicate that a pair of counter-propagating monochromatic waves exhibits opposite phases. This specific phase relationship results in a purely real-valued vector field distribution, as confirmed mathematically by Eqs. (S3) and (S18).

However, this idealized result applies only under lossless conditions. In practice, electromagnetic vector fields experience propagation losses, leading to a complex-

valued field, as investigated and discussed in Ref. 22. In Supplementary Information Section 9, we analyze the topological vector field distribution under such lossy conditions. Our analysis shows that, under weak-loss conditions, the real part of the vector field dominates over the imaginary part, and the topological properties are well preserved. This provides a more comprehensive and rigorous validation of the topological state transitions in optical vector fields.

We sincerely thank the reviewer again for this valuable question, which has helped to enhance the completeness and clarity of our work.

Q5: The authors claim to calculate (and measure) negative topological invariants. I find this to be somewhat confusing, since it seems the two different topological invariants only require that the whole field will have a global phase of π . This, however, cannot uniquely define a field distribution since the global phase is arbitrary. Given this point, all the possible topological invariants will have been spanned within $\beta_1 \leq 2\pi$, instead of within another, more arbitrary number (as in figure 4). It would also mean that the initial transition the authors analyze, not within a Moiré lattice, is no transition at all. The authors should carefully address this point to make sure that their analysis is indeed valid.

R5: We greatly appreciate the reviewer's rigorous comment and the opportunity to elaborate on this important point. The key to resolving the apparent ambiguity regarding the definition of negative topological invariants lies in understanding the time-dependent nature of the electromagnetic vector field and the instantaneous definition of the field skyrmion.

Unlike optical spin skyrmions—which are defined through the time-averaged spin angular momentum and therefore represent stationary field configurations—the field skyrmions discussed in this work are defined from the instantaneous spatial distribution of the electric-field vector. The electromagnetic field is inherently time-varying, and at any given instant, the spatial orientation of the local vector field determines the skyrmion structure at that moment. Consequently, applying a global phase shift to the entire field (e.g., multiplying by $e^{i\phi}$) corresponds physically to observing the field at a different instant in its temporal evolution.

If the vector orientations at two instants are reversed—meaning that every local vector points in the opposite direction—the corresponding skyrmion numbers will take opposite signs (e.g., +1 and -1). These two distributions, while connected through a global temporal phase, are not topologically equivalent, because their real-space vector orientations trace oppositely oriented mappings onto the Poincaré sphere. Therefore, they represent distinct topological configurations of the vector field.

An interesting and relevant example is the work of Giessen *et al.* (*Science* **368**, eaba6415, 2020), which demonstrated a hexagonal field skyrmion lattice—

corresponding to the $\beta = 0$ case in our Fig. 1 with the addition of the full temporal factor $e^{i\omega t}$. In this structure, the field distributions during the first and second halves of an optical cycle become exact inverses of each other, corresponding respectively to skyrmion numbers of $+1$ and -1 . The transition point at $t=\pi/2\omega$ marks a singular configuration where the real-part field amplitude vanishes everywhere, signifying a topological inversion through a null-field state.

Building upon this concept, our work introduces a phase-control parameter β that enables a generalized and continuous realization of such topological inversions at a fixed moment in time. As β varies from 0 to π , the vector field evolves continuously, with its skyrmion number transitioning smoothly from $+1$ to -1 (see Fig. R3). Within each interval— $\beta \in [0, \pi/2)$ and $\beta \in (\pi/2, \pi]$ —the field maintains a well-defined topological invariant ($\text{TI} = +1$ or -1 , respectively), and the transition between them occurs through the emergence of discrete singularities rather than through global field extinction.

Supplementary Fig. R3| Variation of the electric field distribution (E_z) in hexagonal lattices. a-l, Distribution of the out-of-plane electric field component E_z when β varies from 0 (a) to π (g) in intervals of $\pi/6$.

This process provides a physically meaningful and mathematically rigorous route for defining both positive and negative topological invariants in time-varying optical vector fields. It generalizes the previously reported time-driven skyrmion inversions to a broader framework in which the evolution of the field topology is governed by a controllable phase parameter. In this sense, the transitions we report are not artifacts of arbitrary phase choices, but rather represent intrinsic topological transformations that are fully consistent with the vector-field mapping and its associated topological invariant.

Furthermore, we regard the $+1$ and -1 states as distinct topological configurations not only because of their opposite vector-field orientations, but also because the negative skyrmion number plays an essential role in the broader Moiré superlattice framework. In more complex Moiré systems, negative topological invariants naturally emerge as

intermediate states in the continuous evolution of the system's topology. If one acknowledges transitions between two positive topological states within the Moiré framework, it necessarily follows that negative skyrmion numbers must also be included, as they represent indispensable intermediate configurations along the complete topological evolution pathway. Therefore, incorporating both positive and negative skyrmion numbers provides a consistent and comprehensive description of the full range of topological transitions occurring in Moiré superlattices.

Q6: Gerchberg-Saxton phase reconstruction is a tool that should be used very carefully, since it is agnostic of the requirements from Maxwell's equations, as well as from the analytic nature of physical observables. This is particularly problematic in field distributions involving phase singularities, where the GS algorithm exhibits ambiguity. How do the authors make sure their reconstruction does not have artefacts? It is very important that this is addressed, since the entire experimental observation of the authors relies on the reconstruction being accurate.

R6: We appreciate the professional comment from the reviewer. We fully agree that the accuracy of the results recovered by the GS algorithm is crucial for the experiment.

The vector field constructed in our experiment, formed by the interference of TM mode evanescent waves, resembles the one described by Eq. (R1). In such a vector field, whether in a hexagonal lattice or a hexagonal Moiré superlattice, the structure exhibits not only continuous function values but also derivatives of any order. Furthermore, although singularities emerge during topological transitions, these refer specifically to points where the vector direction is undefined, as illustrated in Fig. R3b. Such singularities do not compromise the continuity and smoothness of the vector field itself, and the corresponding phase values remain strictly 0 or $\pm\pi$. Consequently, the GS algorithm converges relatively easily to the target results for this type of structure.

Finally, the inherent robustness of the topological structure helps to mitigate biases introduced during experimental data post-processing. Even in the most ideal case, the experimental outcomes still cannot perfectly match the theoretical predictions. Nevertheless, our research establishes that a necessary condition for the change of topological states is the emergence of new singularities. Therefore, the deviations in the vector field distribution within local regions of the lattice unit cell will not alter the topological properties, provided they do not introduce new singularities. These properties serve as the foundation for the reliable convergence of our experimental results to the target topological state.

Q7: The whole explanation the authors give about the effects of singularities in their system is very confusing. High-symmetry and generic-position singularities are never rigorously defined, so it is not possible to understand whether or not the authors explanation makes sense. I suggest that the authors revise their explanation or exclude it from the final manuscript.

R7: We thank the reviewer for this pertinent comment, and we sincerely apologize for any lack of clarity in our previous description. We have revised the manuscript to provide a clear description of these points. The specific modifications are as follows:

An interesting yet crucial finding in Fig. 2e is that the TIs never take values that are integer multiples of $3/2$. This phenomenon arises from the distribution and classification of singularities, as well as the C_3 symmetry of the structure. The Moiré superlattice described by Eq. (3) possesses C_3 symmetry, where symmetric points are located either at the center or the vertices of the unit cell. Therefore, the center and vertices of the unit cell are termed high-symmetry points, while other positions are termed general-position points. Accordingly, singularities located at high-symmetry points are classified as high-symmetry singularities (red circles in Fig. 3a), whereas those occurring at general positions are regarded as general-position singularities (black circles in Fig. 3a and 3b).

The detailed content can be found in lines 255 to 264 of the main text.

We adopt this classification because the two types of singularities play fundamentally distinct roles in topological transitions. To clarify their respective roles, Supplementary Fig. R4 shows the configurations of systems containing each type of singularity, alongside the corresponding evolution of topological invariants (Supplementary Fig. R4 corresponds to Figs. 3 and 2e in the main text).

Supplementary Fig. R4| Symmetry-Constrained Discretization of Topological Invariants. **a, b,** The unit cells with **(a)** and without **(b)** high-symmetry singularities. **c,** Topological invariants as β_1 and β_2 change independently. High-symmetry singularities and general-position singularities are marked with red and black circles, respectively.

When only general-position singularities are generated (Supplementary Fig. R4b), the topological invariant changes by a multiple of 3. For example, along the process marked by the black dashed arrow in Supplementary Fig. R4c, the invariant changes from -8 to -5 . In contrast, when a high-symmetry singularity is generated (Supplementary Fig. R4a), whether or not accompanied by general-position singularities, the topological

invariant undergoes sign reversal. This is illustrated by the black solid arrow in Supplementary Fig. R4c, where the invariant changes from -8 to $+8$.

In Supplementary Fig. R4c, the maximum attainable value of the topological invariant is 8. Given that the topological invariant can only reverse its sign or change in integer multiples of 3, it follows that its value in Supplementary Fig. R4c cannot be a multiple of 3. Moreover, along the boundaries between adjacent regions in Supplementary Fig. R4c, the topological invariant equals the average of the values on both sides. As a result, none of these boundary values can be integer multiples of $3/2$.

Q8: The authors do not properly cite Ref. 37 (published recently in Nature Physics but under a different name), and also greatly minimize its contribution to the work they present. Whether they relied upon it or not, it is clear that Ref. 37 was the first to describe the majority of the theory the authors show in their work, and while the manuscript does contain many new advances, it does not properly point to the points Ref. 37 has already put forth. Similarly (but to a much smaller extent), the authors do not cite Tsesses et al, Nano Lett. 19 (2019), which observed and discussed some of the field distributions they present (in particular, the honeycomb lattice at $\beta=\pi/2$) or Frischwasser et al, Nat. Photon. 15 (2021), which was – to my knowledge – the first to use the GS algorithm to reproduce the phase of plasmonic fields with distinct topologies.

R8: We thank the reviewer for carefully pointing this out. Indeed, the references mentioned by the reviewer, particularly Ref. 37, are extremely inspiring to the work we reported here. We apologize for missing this and have carefully revised the main text to conduct a more comprehensive discussion of their relevance and specific contributions. The specific modifications made are as follows:

In particular, the Moiré superlattices generated in plasmonic platforms^{40,41} can support arbitrarily large topological invariants. This capability not only significantly enriches the diversity of topological states but also facilitates topological transitions between these states.

Figs. 1b and 1c display the spatial distributions of E_z for $\beta=0$ (Fig. 1b) and $\beta=\pi/2$ (Fig. 1c), respectively, accompanied by the corresponding normalized in-plane electric field vectors. These isolated topological states were previously obtained through methods equivalent to phase modulation⁴⁴. Supplementary Fig. 1 shows the additional E_z field distributions for other β values.

The corresponding E_z -field distributions and reconstructed vector directions, obtained using the Gerchberg–Saxton (GS) algorithm⁴⁷, are shown for three representative values of the encoded parameter: $\beta=0$, $\pi/2$, and π . The GS algorithm is a phase retrieval technique that has been used to reconstruct SPP fields with topological characteristics⁴⁸.

The detailed content can be found in lines 52-55, 123-127, and 311-315 of the main text. (Refs. 37 and 38 in the initial version are now Refs. 40 and 41, respectively, due to structural adjustments).

Q9: The authors should be very clear about their intention when saying they change beta. They write what this means only in the supplementary (i.e., $\varphi_2=\beta/3$, $\varphi_1=\varphi_3=-\beta/3$), but this should be written in the manuscript itself and emphasized, since not every type of change in beta (for example, a change to only one of the phases) will have resulted in the authors' findings.

R9: We thank the reviewer for the thoughtful suggestion. We agree that the quantitative relationship between β and $(\varphi_1, \varphi_2, \varphi_3)$ is fundamental to understanding the existential patterns of the topological invariants. Accordingly, we have moved this analysis from the Supplementary Information to the main text and have emphasized its significance in two key locations. The specific modifications made are as follows:

In the numerical implementation, the relationship between the phase and β can be defined as $\varphi_2 = \beta/3$ and $\varphi_1 = \varphi_3 = -\beta/3$. This choice not only better reflects the symmetry of the structure but also ensures that the center of the unit cell remains at the coordinate origin.

Within the computational framework, the phases are related to the parameters (β_1, β_2) according to the convention of the hexagonal lattice and are selected as follows: $\varphi_{2,1} = \beta_1/3$, $\varphi_{1,1} = \varphi_{3,1} = -\beta_1/3$, and $\varphi_{2,2} = \beta_2/3$, $\varphi_{1,2} = \varphi_{3,2} = -\beta_2/3$.

The detailed content can be found in lines 119-123 and 207-210 of the main text.

Q10: The axes in Figs. 1e, f are unclear, and the analogy to a band structure does not make much sense, since there are no bands in the authors' system. I suggest removing them from the manuscript.

R10: We thank the reviewer for the astute observation. We sincerely apologize for the previously unclear descriptions of the axes in Figs. 1e and 1f. We have revised this section to provide greater clarity for readers in the main text. The specific modifications made are as follows:

During the TTs, the evolution of skyrmions is analogous to that of energy bands. Consequently, the energy band-like structure undergoes a change consistent with that of energy bands: it transitions from a gapped structure (e) to a gapless structure (f) and back to a gapped structure (e).

The detailed content can be found in lines 91 to 94 of the main text.

We certainly agree that the band-like structure proposed in our work shares only a

mathematical formalism with electronic band structures, while their physical origins and interpretations are fundamentally distinct.

We introduced this structure primarily for two reasons. First, it provides a clear and intuitive framework for demonstrating the conditions underlying topological transitions in optical vector fields. Given that topological phase transitions in electronic band structures are well-established, adopting a similar mathematical language within the skyrmion context helps to elucidate the mechanisms driving skyrmion topological transitions. Second, by leveraging this shared mathematical description, we hope to encourage future explorations of deeper connections between skyrmion physics and band theory. We believe this formal analogy may inspire further discoveries of novel physical phenomena at their intersection.

It is for these purposes that we have included the band-like structure in our manuscript. We would be very grateful for any additional suggestions or comments you might have, and we will endeavor to address them thoroughly.

Q11: Although written very well for the most part, there are still some portions with awkward phrasing and typos, including (but not limited to) “the antisymmetric of vector field”, page 8 line 236.

R11: We are grateful for your meticulous review and for identifying the errors in our manuscript. We have conducted a thorough proofreading of the paper and ensured that all minor spelling mistakes are now corrected.

Q12: The authors should add standard deviations on their measured topological invariants in fig. 4.

R12: We thank the reviewer for the valuable suggestion on this. We have revised Fig. 4 in accordance with your recommendation by incorporating error bars that indicate the standard deviation for all applicable data points. This addition provides a more comprehensive representation of the data dispersion and strengthens the statistical validity of our results.

Q13: The rule the authors find for their topological invariants, while very interesting, only applies for C_3 symmetry – i.e., it is not general for other types of symmetry (as is currently written in the manuscript).

R13: Your understanding precisely captures our key finding. In this study, we discovered that the topological states of the vector field in hexagonal Moiré superlattices cannot take integer multiples of $3/2$. The restriction arises from the inherent C_3 symmetry of hexagonal Moiré superlattices. When structural symmetry changes, the existence rules of topological states also change.

To emphasize this fact more clearly, we present topological states in multilayer hexagonal Moiré superlattices and square Moiré superlattices in the Supplementary Information Sections 7 and 8. The results demonstrate that in multilayer hexagonal Moiré superlattices, even with complex structures, the pattern governing the existence of topological invariants remains unchanged due to the preservation of C_3 symmetry. In contrast, the distinct symmetry of square Moiré superlattices leads to a different pattern of topological states, where phase variations result in the topological invariant always remaining zero. This phenomenon arises because the singularities generated in the square Moiré superlattice are typically created in pairs with opposite winding numbers. Consequently, these singularities cancel each other out, leading to no net change in the topological state.

To further clarify that the patterns of topological invariants in our work are specific to certain symmetries, we have emphasized this point in the main text. The specific modifications made are as follows:

Note that the pattern of topological states is governed by the structural symmetry, as demonstrated by our investigation of multilayer Moiré superlattices. In multilayer hexagonal superlattices (Supplementary Information Section 7), the preservation of C_3 symmetry maintains a consistent pattern, even in complex multi-layer configurations. Conversely, the distinct symmetry of the square superlattices (Supplementary Information Section 8) yields a fundamentally different pattern of topological invariants.

The detailed content can be found in lines 294 to 300 of the main text.

I read with great interest the manuscript by B. Tian et al, about implementation of high-order topological invariants in plasmonic fields using Moiré superlattices. Although the idea itself is not new (first performed in Ref. 37 of the manuscript), the authors extend the known theory and understanding of this phenomenon, and showcase control over the topological invariant via an SLM. They characterize the reason for transitions between topological numbers and demonstrate their theory through an experiment.

I find that the authors' idea is innovative, interesting to a broad audience in photonics and topological physics, and – for the most part – performed rigorously (see my comments below for a few “loose-ends”). I am inclined to recommend publication in principle, given that the authors adequately address all comments below:

Major points

- The authors call the transitions in their system “phase transitions”. It is not an accurate phrase, since it is not a many-body system, nor does it contain any form of nonlinearity. The cited works in topological photonics certainly do not show phase transitions, but rather – just “topological transitions”, which are the commonly agreed upon term. I request that the authors amend their manuscript in accordance with this point.
- The claim of showing topological invariants between -58 and 58 is unsubstantiated. The authors show this in theory, while measuring only -8 to 8. I suggest that either the authors amend their claim, or properly measure topological invariants up to 58.
- I greatly appreciate that the authors check whether or not their minimal field amplitudes tend to 0. In such points, the topological density is ill-defined, so the whole distribution cannot have a well-defined topological charge other than 0. In this respect, any fractional topological charge the authors measure is probably occurring in such a point, and therefore – is ill-defined. In addition, is it true that the dashed lines in fig. 2e coincide with zero minimal field amplitude lines shown in fig. 2f? this would mean that, along these lines, the topological constant is ill-defined, whereas right now it is defined via a particular color in fig. 2e. Therefore, the figure may have to change, and the authors should complete their check that indeed the topological density in cases they measure is well-defined.
- In the same vein of checking that the distribution is well-defined, how do the authors make sure that the real part of the electric field is indeed much larger than the imaginary part? Of course, this can be defined up to a global phase, but to take only one part of the field, it must to be much larger than the other part, with the criterion defined in the supplementary information of Ref. 22. It seems, from looking at the supplementary, that this might not be the case for some of the measurements. I strongly

recommend that the authors check this point to make sure that their main claims are substantiated.

- The authors claim to calculate (and measure) negative topological invariants. I find this to be somewhat confusing, since it seems the two different topological invariants only require that the whole field will have a global phase of π . This, however, cannot uniquely define a field distribution since the global phase is arbitrary. Given this point, all the possible topological invariants will have been spanned within $\beta_1 \leq 2\pi$, instead of within another, more arbitrary number (as in figure 4). It would also mean that the initial transition the authors analyze, not within a Moiré lattice, is no transition at all. The authors should carefully address this point to make sure that their analysis is indeed valid.
- Gerchberg-Saxton phase reconstruction is a tool that should be used very carefully, since it is agnostic of the requirements from Maxwell's equations, as well as from the analytic nature of physical observables. This is particularly problematic in field distributions involving phase singularities, where the GS algorithm exhibits ambiguity. How do the authors make sure their reconstruction does not have artefacts? It is very important that this is addressed, since the entire experimental observation of the authors relies on the reconstruction being accurate.
- The whole explanation the authors give about the effects of singularities in their system is very confusing. High-symmetry and generic-position singularities are never rigorously defined, so it is not possible to understand whether or not the authors explanation makes sense. I suggest that the authors revise their explanation or exclude it from the final manuscript.

Minor points

- The authors do not properly cite Ref. 37 (published recently in Nature Physics but under a different name), and also greatly minimize its contribution to the work they present. Whether they relied upon it or not, it is clear that Ref. 37 was the first to describe the majority of the theory the authors show in their work, and while the manuscript does contain many new advances, it does not properly point to the points Ref. 37 has already put forth. Similarly (but to a much smaller extent), the authors do not cite Tsesses et al, Nano Lett. 19 (2019), which observed and discussed some of the field distributions they present (in particular, the honeycomb lattice at $\beta = \pi/2$) or Frischwasser et al, Nat. Photon. 15 (2021), which was – to my knowledge – the first to use the GS algorithm to reproduce the phase of plasmonic fields with distinct topologies.
- The authors should be very clear about their intention when saying they change β . They write what this means only in the supplementary (i.e., $\phi_2 = \beta/3, \phi_1 = \phi_3 = -\beta/3$), but this should be written in the manuscript itself and emphasized, since not

every type of change in beta (for example, a change to only one of the phases) will have resulted in the authors' findings.

- The axes in fig. 1e,f are unclear, and the analogy to a band structure does not make much sense, since there are no bands in the authors' system. I suggest removing them from the manuscript.
- Although written very well for the most part, there are still some portions with awkward phrasing and typos, including (but not limited to) "the antisymmetric of vector field", page 8 line 236.
- The authors should add standard deviations on their measured topological invariants in fig. 4.
- The rule the authors find for their topological invariants, while very interesting, only applies for C3 symmetry – i.e., it is not general for other types of symmetry (as is currently written in the manuscript).